# Mechanism of substrate binding and transport in BASS transporters

Patrick Becker[1], Fiona Naughton[2†], Deborah Brotherton[1], Raul Pacheco-Gomez[3], Oliver Beckstein[2]*, Alexander D Cameron[1]*

[1]School of Life Sciences, University of Warwick, Coventry, United Kingdom; [2]Department of Physics, Arizona State University, Tempe, United States; [3]Malvern Panalytical Ltd, Malvern, United Kingdom

*For correspondence:
obeckste@asu.edu (OB);
a.cameron@warwick.ac.uk (ADC)

Present address: †Cardiovascular Research Institute, University of California San Francisco, San Francisco, United States

**Abstract** The bile acid sodium symporter (BASS) family transports a wide array of molecules across membranes, including bile acids in humans, and small metabolites in plants. These transporters, many of which are sodium-coupled, have been shown to use an elevator mechanism of transport, but exactly how substrate binding is coupled to sodium ion binding and transport is not clear. Here, we solve the crystal structure at 2.3 Å of a transporter from *Neisseria meningitidis* (ASBT$_{NM}$) in complex with pantoate, a potential substrate of ASBT$_{NM}$. The BASS family is characterised by two helices that cross-over in the centre of the protein in an arrangement that is intricately held together by two sodium ions. We observe that the pantoate binds, specifically, between the N-termini of two of the opposing helices in this cross-over region. During molecular dynamics simulations the pantoate remains in this position when sodium ions are present but is more mobile in their absence. Comparison of structures in the presence and absence of pantoate demonstrates that pantoate elicits a conformational change in one of the cross-over helices. This modifies the interface between the two domains that move relative to one another to elicit the elevator mechanism. These results have implications, not only for ASBT$_{NM}$ but for the BASS family as a whole and indeed other transporters that work through the elevator mechanism.

## eLife assessment

The manuscript represents an **important** contribution to an ongoing discussion about the substrate binding site and mechanism of the Bile Acid Sodium Symporter (BASS) family of transporters. Structural and biochemical analysis of a bacterial homolog, ASTBnm, in complex with its native substrate (not bile acids, but a vitamin A precursor, pantoate) show a new binding site that is consistent with classical proposals for elevator-type transport mechanisms. Molecular dynamics (MD) simulations highlight the improved stability for the substrate in the active site when ions are present, suggesting a binding order during the transport cycle. The structural studies, binding assays, and MD simulations are **convincing**.

## Introduction

The bile acid sodium symporter (BASS) family of secondary transporters is synonymous with its founding members, the apical sodium-dependent bile acid transporter (ASBT) and the sodium taurocholate cotransporting polypeptide (NTCP) (*Geyer et al., 2006*). These proteins harness the sodium ion gradient to transport bile acids across the plasma membranes of enterocytes of the terminal ileum and hepatocytes, respectively. They are both targets of drugs currently in the clinic; ASBT as the target of drugs to alleviate chronic constipation (*Karpen et al., 2020*; *Khanna and Camilleri, 2021*) and NTCP as a target for hepatitis B and D virus entry inhibitors (*Wedemeyer et al., 2023*). Both proteins

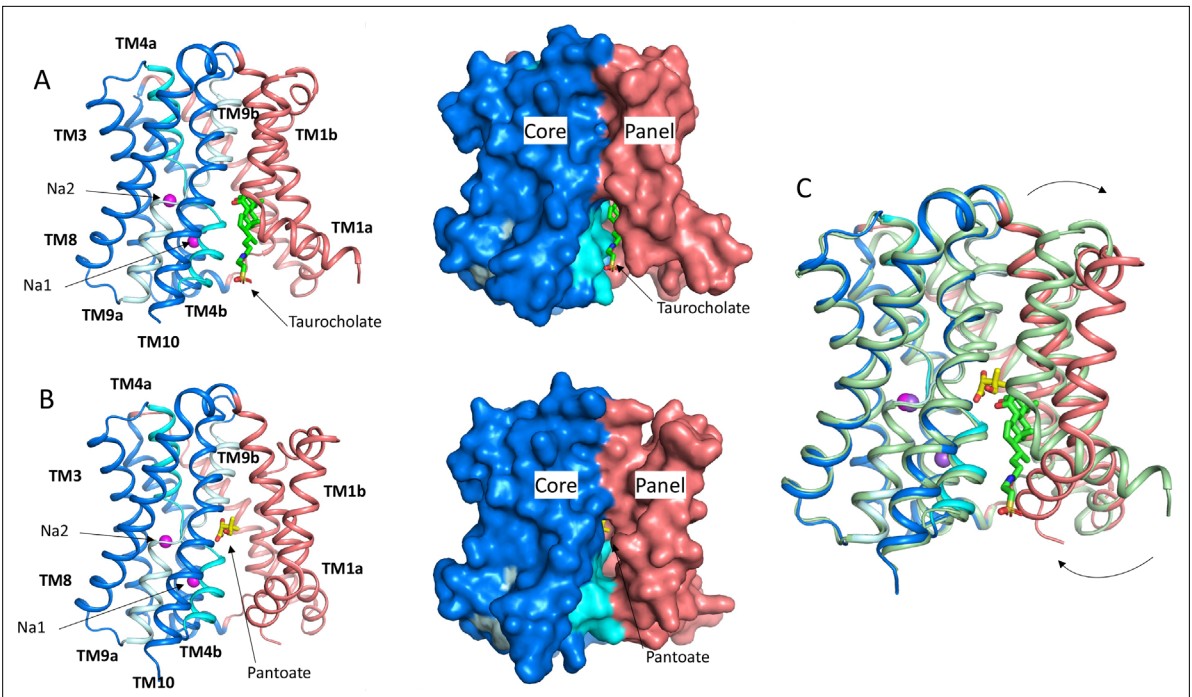

**Figure 1.** Structure of ASBT$_{NM}$. (**A**) Structure of ASBT$_{NM}$ in complex with taurocholate ASBT$_{NM(TCH)}$ (*Hu et al., 2011*). The panel domain is coloured salmon. The core domain is coloured blue with the cross-over helices, TM4 and TM9 in cyan and pale blue, respectively. The taurocholate is shown in a stick representation with green carbon atoms and the sodium ions are shown as magenta spheres. Left, cartoon representation, Right, surface representation. (**B**) As A for the structure in complex with pantoate (ASBT$_{NM(Pan)}$). The pantoate is depicted with yellow carbon atoms. (**C**) Superposition of ASBT$_{NM(Pan)}$ (colouring as A) on ASBT$_{NM(TCH)}$ (pale green carbon atoms). The arrows show the movement of TM1.

The online version of this article includes the following video and figure supplement(s) for figure 1:

**Figure supplement 1.** Sequence alignment.

**Figure 1—video 1.** Morph between ASBT$_{NM(Pan)}$ and ASBT$_{NM(TCH)}$.
https://elifesciences.org/articles/89167/figures#fig1video1

also influence drug distribution. The BASS family, however, transports a wide array of substrates other than bile acids. In mammals, the sodium-dependent organic anion transporter (SOAT) transports sulphated steroids (*Grosser et al., 2013*) and others are putative neurotransmitter transporters (*Burger et al., 2011*). In plants sodium-coupled BASS transporters transport small metabolites such as pyruvate (*Furumoto et al., 2011*) and glycolate (*South et al., 2017*) across the plastidial membrane.

The first detailed structural information on the BASS transporters came through crystal structures of two bacterial transporters, one from *Neisseria meningitidis* (ASBT$_{NM}$) (*Hu et al., 2011*) and one from *Yersinia frederiksinii* (ASBT$_{YF}$) (*Zhou et al., 2014*). Though neither protein is likely to transport bile acids physiologically, in vitro both transporters have been shown to catalyse the sodium-dependent transport of the bile acid taurocholate (TCH) and have provided an initial structural framework through which the extensive site-directed mutagenesis studies carried out on ASBT and NTCP could be mapped. The bacterial transporters are built from 10 transmembrane helices, with a twofold inverted repeat arranged in two domains, a core domain, and a panel domain (*Figure 1A*). The core domain is characterised by two extended helices that cross over at the centre with residues within the extended region contributing to two sodium ion binding sites (Na1 and Na2). The residues forming the sodium binding site in ASBT$_{NM}$ are conserved in many members of the BASS family including ASBT and NTCP (*Figure 1—figure supplement 1*). ASBT$_{NM}$ was crystallised in the presence of TCH and this bile acid is observed to bind in an inward-facing cavity between the core and panel domains in a binding mode that remains stable during molecular dynamics (MD) simulations (*Alhadeff et al., 2015*). Secondary transporters function by the alternating access mechanism in which conformational changes to the transporter enable the substrate binding site to switch between the opposing sides of the membrane (*Beckstein and Naughton, 2022*; *Drew and Boudker, 2016*). The structure of ASBT$_{YF}$ was solved in

both an inward-facing state, similar to ASBT$_{NM}$, and an outward-facing state. Based on these structures an elevator-type model of transport was proposed (*Zhou et al., 2014*). In such mechanisms it is expected that the substrate binds to one domain, which moves with respect to another so that the substrate can be carried across the membrane (*Drew and Boudker, 2016*). In the ASBT$_{NM}$ structure, however, the position of the TCH is not entirely consistent with such a model, as though there are specific interactions only with residues of the core domain, the TCH is not primarily embedded within that domain and is not set as deeply within the cleft as might be expected. This may be partly due to the protein binding to the inward-facing state of the protein where the substrate should be released, but it is also likely that the bile acids do not bind optimally to the bacterial transporters. More recently, structures of human NTCP have been reported (*Asami et al., 2022*; *Goutam et al., 2022*; *Liu et al., 2022*; *Park et al., 2022*). Though NTCP lacks the first transmembrane helix of the bacterial transporters, the overall fold of the proteins is the same and similar conformations of outward- and inward-facing states consistent with an elevator mechanism of transport are observed (*Park et al., 2022*). The structure of NTCP was also solved with glyco-chenodeoxycholic (*Liu et al., 2022*). In this structure two molecules of the bile acid are observed binding to the protein with weak interactions to the core domain. The unusual binding modes led the authors of this paper to propose an alternative to the classical alternating access mechanism that involves the binding of two substrates, only one of which is transported in each cycle. To gain further insight into the mechanism of BASS family of transporters, and in particular the 10-transmembrane helix transporters, we therefore sought to find a likely substrate for the bacterial transporters that would enable us to understand how substrates bind.

ASBT$_{NM}$ and ASBT$_{YF}$ have high sequence identity to PanS from *Salmonella enterica* (43% and 83% sequence identity respectively) (*Figure 1—figure supplement 1*). PanS has been implicated in the transport of the coenzyme A precursors, ketopantoate, and pantoate (*Ernst and Downs, 2015*). ASBT$_{NM}$ and ASBT$_{YF}$ are also similar to BASS1 (*Figure 1—figure supplement 1*) and it has been shown that BASS1 from *A. thaliana* can transport pantoate, at least in vitro (*Huang et al., 2018*). We therefore decided to investigate whether the coenzyme A precursors would also bind to ASBT$_{NM}$.

Here, we demonstrate that pantoate, but not ketopantoate or pantothenate, binds to ASBT$_{NM}$. We solve the crystal structure of the protein in complex with pantoate and show that the pantoate makes specific interactions with residues in the cross-over region of the protein consistent with the elevator mechanism of transport. MD shows that this binding mode is more stable when sodium ions are present in their respective binding sites. Binding of pantoate causes a subtle conformational change within the core region of the protein, which may trigger the more widespread movements of the protein that would enable transport to occur. This suggests a more specific mechanism for ASBT$_{NM}$,

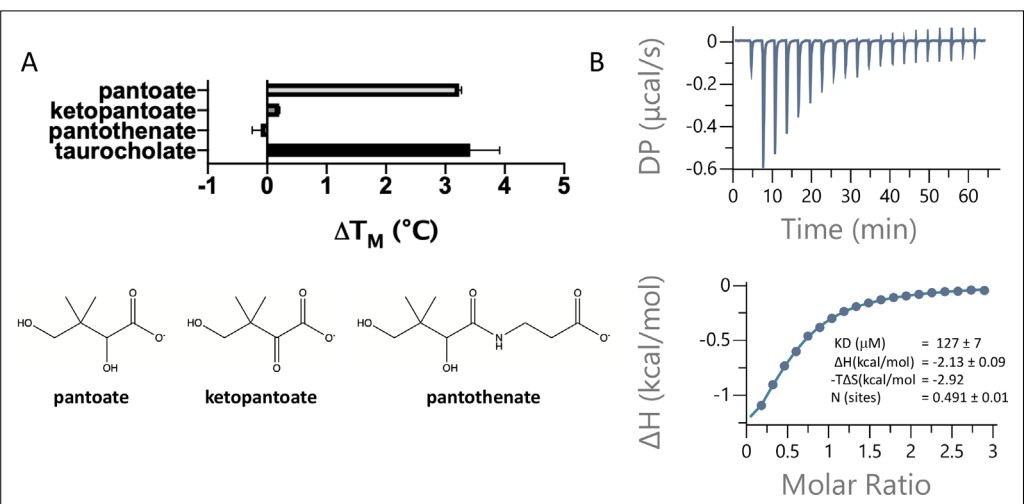

**Figure 2.** Pantoate binding to ASBT$_{NM}$. (**A**) Results from thermostability assay showing that pantoate stabilises ASBT$_{NM}$ to a similar extent to taurocholate. The compounds are shown below. The mean and standard deviations are shown based on three individual experiments. (**B**) Pantoate binding to ASBT$_{NM}$ measured by isothermal calorimetry.

much more in line with the classical alternating access model of transport, than has recently been suggested for NTCP.

## Results

### Pantoate binds to ASBT$_{NM}$

To assess whether pantoate and its derivatives are likely substrates for ASBT$_{NM}$, we first used a dye-based stability assay in which stability is used as a surrogate for binding (*Alexandrov et al., 2008*). Whereas pantoate stabilised the protein to a similar amount to TCH, neither ketopantoate nor pantothenate had any effect under the conditions tested (*Figure 2a*). To verify binding and obtain a more reliable estimate of the affinity of pantoate for ASBT$_{NM}$ isothermal calorimetry (ITC) was then used, giving a measured $K_D$ of 127 µM (*Figure 2b*). Although we were unable to obtain a reliable estimate of the $K_D$ of TCH using ITC due to its detergent-like properties, the $K_D$ of pantoate is similar to the $K_M$ reported for TCH (50 µM) (*Hu et al., 2011*).

### Structure of ASBT$_{NM}$ with pantoate

To understand how pantoate binds to ASBT$_{NM}$ we solved the structure of the protein in the presence of pantoate using X-ray crystallography (ASBT$_{NM(Pan)}$). Crystals were grown using the in meso method of crystallisation and the structure was solved by molecular replacement and refined at a resolution of 2.3 Å (*Table 1*). Density consistent with pantoate, and sodium ions in both ion binding sites is evident in the resulting maps (*Figure 3B*, *Figure 3—figure supplement 1*). There is also evidence of a lipid-like molecule within the binding site. The transporter adopts an inward-facing conformation as seen for the structure of ASBT$_{NM}$ with TCH present (ASBT$_{NM(TCH)}$; 3ZUX) (*Hu et al., 2011*) and the two structures can be superposed with a root mean square deviation (RMSD) of 0.6 Å for 263 out of 308 C$_\alpha$ atoms within 2 Å after superposition (see Materials and methods). The most substantial difference in the conformation of the two structures is seen for TM1. In the ASBT$_{NM(TCH)}$ structure, TM1 bounds one side of the crevice between the panel and core domains (*Figure 1A*). In this structure the helix is kinked at residue Thr 14 so that it splays out and enlarges the cavity on the inward-facing side of the protein (*Figure 1A*). In the ASBT$_{NM(Pan)}$ structure TM1 is still kinked although the whole helix has moved as an approximate rigid body by ~75° pivoting around Ile 11 such that residues 1–10 move over the cytoplasmic entrance to the cavity to partially occlude it from the inward-facing side, and residues 12–28 move away from TM10. This creates an opening into the crevice from the membrane between the panel and core domains (*Figure 1B and C*, *Figure 1—video 1*).

The pantoate binds between the two cross-over helices TM4b and TM9b (*Figure 3*). The carboxylic acid of the pantoate interacts with the main chain nitrogen atoms of Thr 112 and Ala 113 of TM4b and the 2-hydroxyl oxygen is within hydrogen bonding distance of the main chain nitrogen of Gly 267 of TM9b and the amino oxygen of Asn 265. The hydroxyl oxygen of the methyl-propanol moiety also interacts with His 294 and Asn 295, which reside on TM10. These residues are all within the core domain. The closest residues to the pantoate on the panel domain are Ile 203 and Ile 47 that interact with the methyl-propanol moiety. The sodium ion binding sites in ASBT$_{NM}$ are also located at the cross-over region of the two helices, behind the pantoate when viewed from the crevice between the core and panel domains (*Figure 3C*). The ions are clearly defined in the electron density (*Figure 3—figure supplement 1A*) and there is very little change in their coordination in the ASBT$_{NM(Pan)}$ structure with respect to that of ASBT$_{NM(TCH)}$ (*Figure 3—figure supplement 1B*).

Within the region of the sodium and pantoate binding sites the most obvious change in the pantoate-bound structure relative to ASBT$_{NM(TCH)}$ is that the main chain nitrogen of Thr 112 is displaced by ~1 Å and the C$_\gamma$ by 2.4 Å (*Figure 3D*). In fact, there is a slight movement of the whole of TM4b, which includes the sodium ion ligands, Ser 114 and Asn 115, towards the pantoate (*Figure 3D*, *Figure 1—video 1*). On the panel domain Ile 203, located at the centre of TM7, is also displaced slightly (~1.2 Å) (*Figure 3D and E*) enabling the pantoate to be accommodated easily. The conformational change of TM1 may be triggered by this displacement given that Ile 203 would clash with Phe 15 if TM1 had adopted the same conformation as in ASBT$_{NM(TCH)}$ (*Figure 3E*).

**Table 1.** Data processing and refinement statistics.

| | ASBT$_{NM(Pan)}$ | ASBT$_{NM(ns)}$ |
|---|---|---|
| Wavelength (Å) | 0.9999 | 0.9999 |
| Resolution range | 43.78–2.3 (2.382–2.3) | 58.96–2.1 (2.175–2.1)* |
| Space group | C2 | P 2$_1$ 2$_1$ 2$_1$ |
| Unit cell: a, b, c (Å),α, β, γ (°) | 85.0 89.4 53.1 90 124.4 90 | 49.5 80.6 86.5 90 90 90 |
| Total reflections | 55,444 (3034) | 146,457 (14,741) |
| Unique reflections | 14,273 (1201) | 20,744 (2032) |
| Multiplicity | 3.9 (2.5) | 7.1 (7.3) |
| Completeness (%) | 97.7 (82.8) | 99.4 (99.3) |
| Mean I/sigma(I) | 5.7 (1.7) | 6.7 (2.0) |
| Wilson B-factor | 36 | 30 |
| R-merge | 0.1181 (0.4632) | 0.1356 (0.8541) |
| R-meas | 0.1368 (0.5779) | 0.1467 (0.9198) |
| R-pim | 0.06752 (0.3397) | 0.05402 (0.3316) |
| CC1/2 | 0.988 (0.804) | 0.957 (0.658) |
| CC* | 0.997 (0.944) | 0.989 (0.891) |
| Reflections used in refinement | 14,259 (1201) | 20,691 (2027) |
| Reflections used for R-free | 774 (55) | 1089 (96) |
| R-work | 0.2284 (0.3488) | 0.2115 (0.2933) |
| R-free | 0.2648 (0.4564) | 0.2387 (0.2885) |
| CC(work) | 0.937 (0.861) | 0.929 (0.846) |
| CC(free) | 0.927 (0.678) | 0.835 (0.907) |
| Number of non-hydrogen atoms | 2324 | 2412 |
| Macromolecules | 2276 | 2282 |
| Ligands | 34 | 135 |
| Solvent | 25 | 34 |
| Protein residues | 310 | 310 |
| RMS (bonds) | 0.002 | 0.002 |
| RMS (angles) | 0.44 | 0.53 |
| Ramachandran favoured (%) | 98.05 | 99.03 |
| Ramachandran allowed (%) | 1.95 | 0.97 |
| Ramachandran outliers (%) | 0 | 0 |
| Rotamer outliers (%) | 0.85 | 0.84 |
| Clashscore | 2.52 | 3.25 |
| Average B-factor | 52.2 | 41.5 |
| Macromolecules | 52.3 | 40.3 |
| Ligands | 46.0 | 69.4 |
| Solvent | 47.6 | 42.8 |
| Number of TLS groups | 1 | 1 |

*Statistics for the highest resolution shell are shown in parentheses.

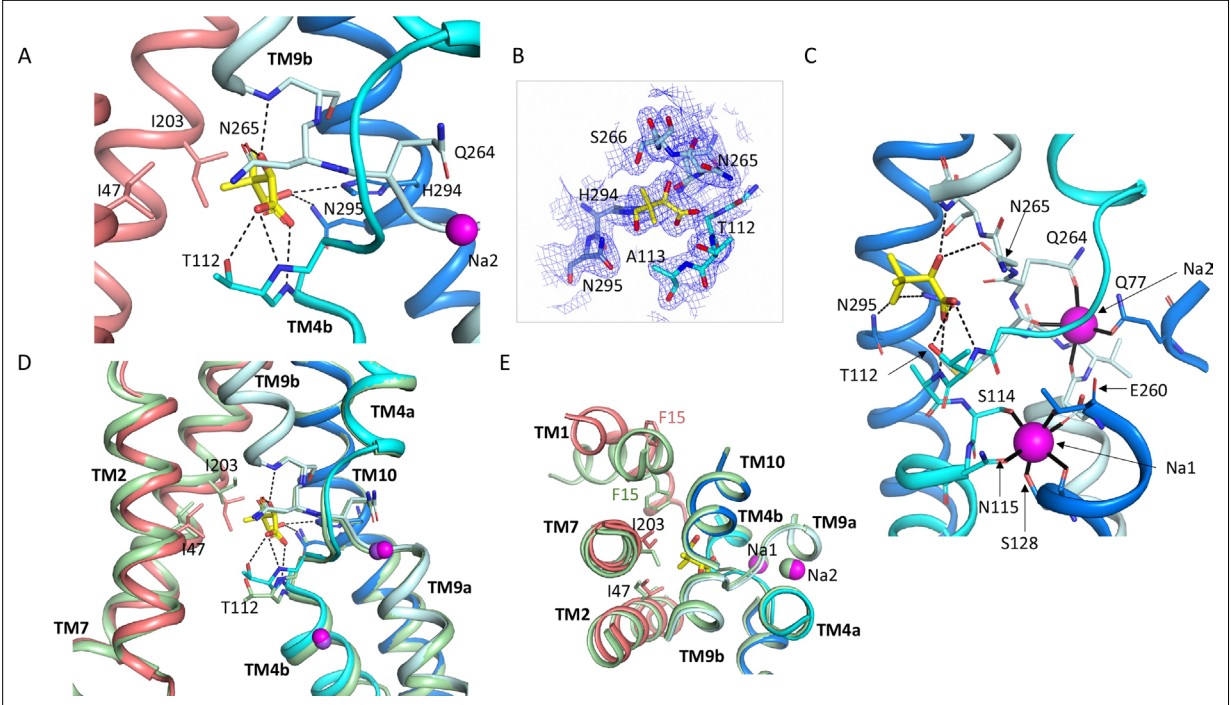

**Figure 3.** Pantoate binding site. (**A**) The pantoate binding site in the ASBT$_{NM(Pan)}$ structure, coloured as in *Figure 1*. Hydrogen bonds are shown as dashed lines. (**B**) 2mFo-Fc density for the refined structure. The density is contoured at 1σ. (**C**) View of the ASBT$_{NM(Pan)}$ structure highlighting the juxtaposition of the residues interacting with the sodium ions and those interacting with the pantoate. (**D**) Superposition of the ASBT$_{NM(TCH)}$ structure (pale green) on the ASBT$_{NM(Pan)}$ structure highlighting the difference in position of TM4b and especially Thr 112 between the two structures. (**E**) As D but shown from the extracellular side highlighting the differences in position of Ile 203 and Phe 15.

The online version of this article includes the following figure supplement(s) for figure 3:

**Figure supplement 1.** Sodium site for the ASBT$_{NM(Pan)}$ structure.

## Structure of ASBT$_{NM}$ without substrate

Given that the subtle conformational changes between the pantoate and TCH-bound structures would be consistent with mechanistic changes upon substrate binding, with ASBT$_{NM(TCH)}$ representing a non-substrate bound structure, we also solved the structure without TCH or pantoate present (ASBT$_{NM(ns)}$) at 2.1 Å using the in meso method of crystallisation (*Table 1*). Overall, ASBT$_{NM(ns)}$ is very similar to ASBT$_{NM(TCH)}$ with an RMSD of 0.5 Å for 293 out of 308 C$_\alpha$ atoms (see Materials and methods) and an almost identical coordination of the sodium ions (*Figure 4*, *Figure 4—figure supplement 1*). There are only two regions where there are slightly larger changes. The first, again, centres on TM1. However, the change is much more subtle than to the pantoate-bound structure, pivoting ~15° at Thr 14 (*Figure 4A and B*). The second is in the loop between TM5 and TM6, which links the core to the panel domain where the loop takes a conformation more similar to that seen in the ASBT$_{NM(Pan)}$ structure. In flexing between the inward- and outward-facing structures, as reported for ASBT$_{YF}$ (*Zhou et al., 2014*), this loop changes conformation as it allows the panel to move with respect to the core. Overall, therefore, it appears that the binding of pantoate, rather than either the absence of TCH or the difference in crystallisation method, causes the change in position of TM4b.

## Specificity of pantoate binding

To probe the specificity of binding we used two approaches. Firstly, we mutated the two residues for which the side chains are within hydrogen bonding distance of the pantoate in the ASBT$_{NM(Pan)}$ structure and tested the affinity of pantoate for the resultant proteins by ITC. Whereas the mutation of Asn 265 to alanine caused the binding of pantoate to be abolished (*Figure 5A*), mutation of Thr 112 to either valine or alanine surprisingly resulted in an increase in affinity to 87 µM (*Figure 5B*) and 11 µM (*Figure 5C*) respectively, though noticeably the latter was entropy driven. Secondly, because much of the molecular recognition involves the main chain atoms, we used a structure-activity relationship

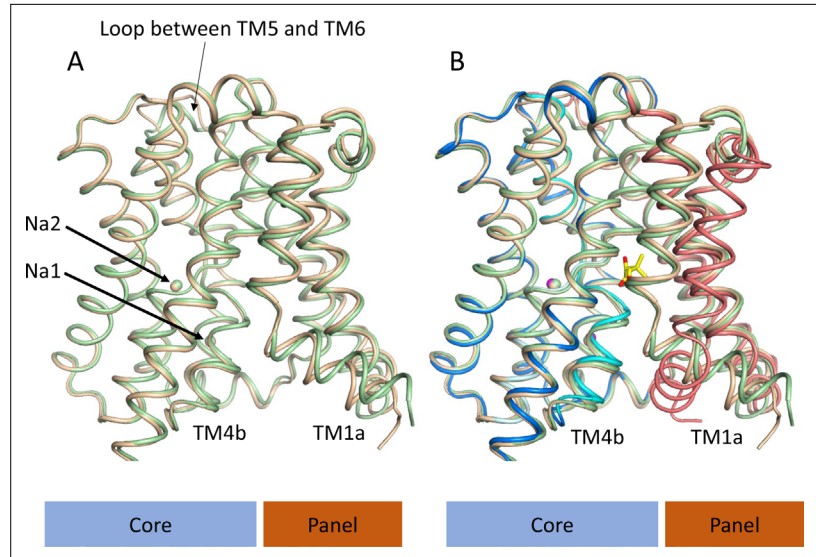

**Figure 4.** Structure of ASBT$_{NM}$ without pantoate or taurocholate. (**A**) Superposition of ASBT$_{NM(ns)}$ (wheat) on ASBT$_{NM(TCH)}$ (pale green) highlighting the similarity of the two structures. The main differences are in the position of TM1, where TM1a adopts a slightly different angle with respect to TM1b, and in the loop between TMs 5 and 6, which links the core domain to the panel domain. (**B**) As A with the addition of the ASBT$_{NM(Pan)}$ structure (coloured as in *Figure 1*). The difference in the position of TM1 and TM4b, with respect to the two structures without pantoate is evident.

The online version of this article includes the following figure supplement(s) for figure 4:

**Figure supplement 1.** Electron density associated with ASBT$_{NM(ns)}$.

approach, testing whether a panel of similar compounds would stabilise the protein (*Figure 5—figure supplement 1*). None of the compounds tested stabilised the protein as much as pantoate, showing the importance of the hydroxy-acetate group, which interacts with the main chain atoms. Just replacing the hydroxyl oxygen with a ketone as in ketopantoate appears to disrupt binding, likely due to an unfavourable interaction with Asn 265. The only other residues that possess the hydroxy-acetate moiety are isocitrate and D-malate. Both compounds have additional charged groups that may make them less favourable for binding.

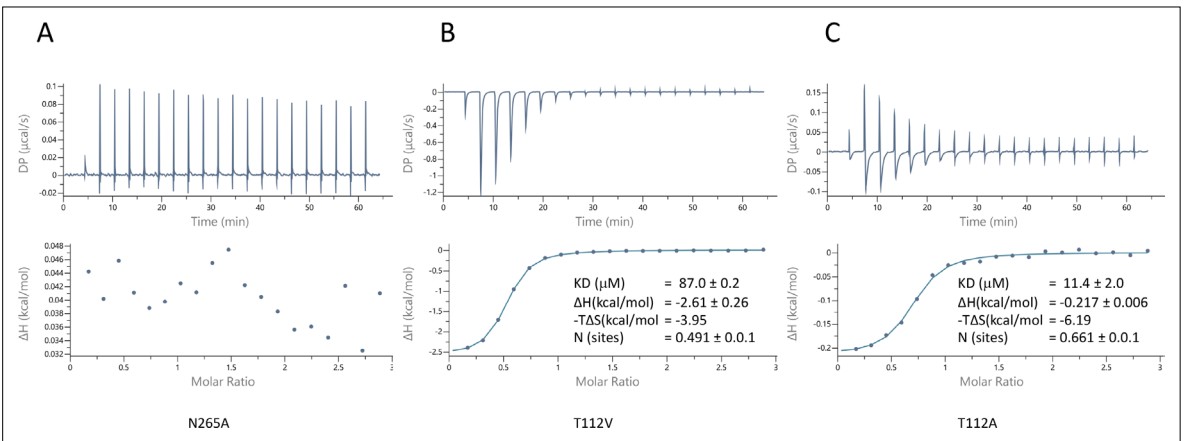

**Figure 5.** Characterisation of pantoate binding to mutants of ASBT$_{NM}$. Pantoate binding to ASBT$_{NM}$ mutants measured by isothermal calorimetry.

The online version of this article includes the following figure supplement(s) for figure 5:

**Figure supplement 1.** Testing a panel of compounds for potential binding to ASBT$_{NM}$.

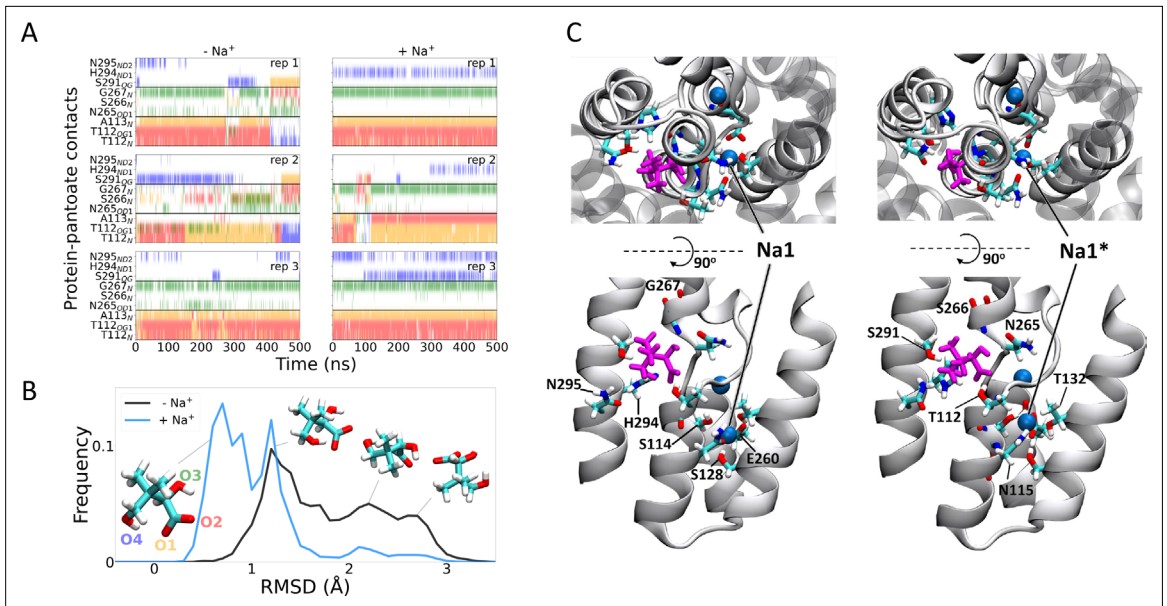

**Figure 6.** Molecular dynamics simulations. (**A**) Hydrogen bonds between the pantoate and protein followed over the course of the simulations starting without (left) or with (right) sodium bound. Red, yellow, green, and blue indicate a contact with the O1, O2, O3, and O4 atoms of pantoate, respectively (as shown in B). Contacts are shown for all residues with contacts in greater than 10% of any simulation. (**B**) Histograms of pantoate heavy-atom root mean square deviation (RMSD) over three simulations starting with (blue) or without (black) bound sodium, calculated following $C_\alpha$ alignment of the protein around the binding site (residues 108–117 [TM4], 199–207 [TM9], and 287–296 [TM10]). Representative snapshots of pantoate, relative to the starting position on the left are shown. (**C**) Representative snapshots showing bound pantoate (magenta) and sodium (blue spheres), showing the location of the canonical sodium binding site 1 and the alternate site 1*. ASBT is shown with cartoon representation; for clarity, only helices contributing to the binding sites (TMs 4, 5, 9, 10) are shown in the lower (side view) panels. Residues making up the pantoate and Na1 binding sites are shown in stick representation.

The online version of this article includes the following figure supplement(s) for figure 6:

**Figure supplement 1.** Root mean square deviation (RMSD) of protein and pantoate during simulations.

**Figure supplement 2.** Sodium binding during simulations.

## Molecular simulations show greater stability of pantoate when sodium ions are present

To gain insight into the effect of sodium ions on the binding of pantoate, MD simulations were carried out. Over 0.5 µs of MD simulations, pantoate remains in the crystallographic binding position (*Figure 6*). With sodium ions present in both the Na1 and Na2 sites, the hydrogen bonds between pantoate and the main chain nitrogen atoms of T112 and A113 at the N-terminus of TM4b and G267 at the N-terminus of TM9b remain intact (*Figure 6A*) with more fluctuating interactions with the side chain atoms. On the other hand, in the absence of sodium ions the pantoate is more mobile (*Figure 6B*, *Figure 6—figure supplement 1A*) with the interactions with the main chain atoms more intermittent (*Figure 6A*). Over the course of the simulations with the sodium ions, the ions remain stably bound in or close to the Na1 and Na2 sites although the simulations indicate that there is an alternative sodium ion binding position close to the crystallographic Na1 site (labelled Na1* in *Figure 6C*). In the simulations without bound sodium ions, ions enter the inward-facing funnel and approach the Na1* binding site, but do not settle into the same binding mode seen in the bound simulations (*Figure 6—figure supplement 2*), although complete binding events may occur on longer time scales.

## Discussion

ASBT$_{NM}$ binds pantoate, consistent with this compound being suggested as a substrate for the homologous PanS (*Ernst and Downs, 2015*) and BASS1 (*Huang et al., 2018*) proteins. The results from the MD simulations demonstrate the importance of the sodium ions in stabilising the binding mode of the pantoate at the N-termini of the cross-over helices (*Figure 6*) and might suggest that the ions

structure the region in readiness for substrate binding. This conclusion is supported by the sodium-free wild-type structures of ASBT$_{YF}$ (*Wang et al., 2021*; *Zhou et al., 2014*), where the region equivalent to 110–116 in ASBT$_{NM}$ and the pantoate interacting residues Thr 112 and Ala 113 adopt varying positions. While it would appear from the MD simulations that the interaction that the pantoate makes with the side chain of Asn 265 is less conserved than those involving the main chain atoms, the mutagenesis studies highlight the importance of the residue in binding. Asn 265 also caps TM4b so could potentially stabilise the structure of the binding site as well as interacting with the substrate.

The pantoate is firmly nestled within the core domain and the only interactions it makes with the panel domain are van der Waals interactions with Ile 47 and Ile 203. These residues are conserved in PanS and ASBT$_{YF}$ but in BASS1 are replaced with a valine and a threonine respectively (*Figure 1—figure supplement 1*). The binding of the pantoate appears to cause a displacement of Ile 203, which in turn displaces Phe 15 on TM1. The novel position of TM1 appears to partly occlude the pantoate in the binding site. It would be tempting to think that the partially occluded conformation we observe here is mechanistic as transporters often go through one or more occluded conformations during their mechanistic cycle (*Beckstein and Naughton, 2022*). However, while Phe 15 is conserved in PanS, ASTB$_{YF}$, and BASS1, and there is some flexing of this region in ASBT$_{YF}$ as the protein changes conformation from outward- to inward-facing (*Zhou et al., 2014*), there is little conservation at the N-terminus amongst the proteins, suggesting that the position of the N-terminus may not be critical for transport.

*Zhou et al., 2014* have demonstrated that ASBT$_{YF}$ is likely to go through an elevator mechanism as is also observed for other proteins with the same fold (*Fang et al., 2021*; *Lee et al., 2013*; *Park et al., 2022*; *Ung et al., 2022*). As a sodium-coupled symporter, the pantoate should bind to the outward-facing form of the protein and trigger movement to the inward-facing state where it can be released. Modelling the outward-facing state of the ASBT$_{NM(Pan)}$ structure based on the ASBT$_{YF}$ structure shows that the pantoate could easily be accommodated in the outward-facing form (*Figure 7—figure supplement 1*). The position of the pantoate observed here, therefore, is consistent with an elevator mechanism. It is possible that with the constraints of the membrane, the interaction of Ile 207

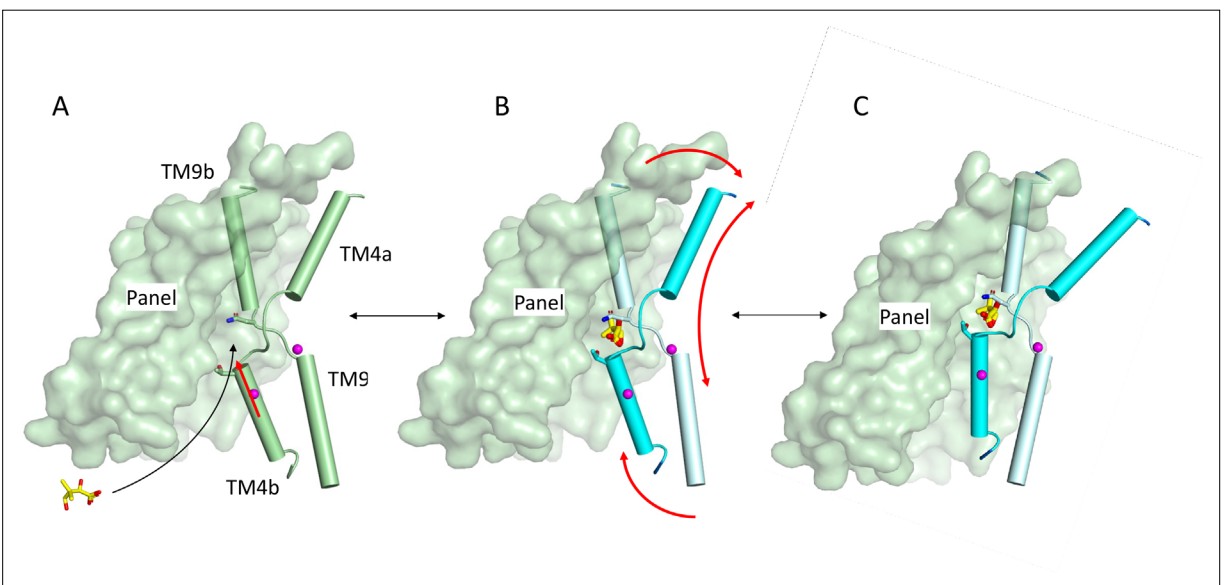

**Figure 7.** Schematic of mechanism. Pantoate binding to the cross-over region between TM4b and TM9b of the substrate-free structure (**A**) elicits a conformational change in TM4b (red arrow) (**B**). The change in conformation of the core region of the protein may allow greater freedom of movement of the panel domain relative to the core enabling it to swing upwards in an elevator movement (**C**) (red arrows in B). The position of the core relative to the panel domain in **C** was based on the relative positions of the two domains in the outward-facing structure of ASBT$_{YF}$.

The online version of this article includes the following video and figure supplement(s) for figure 7:

**Figure supplement 1.** Pantoate binding to an outward-facing state model.

**Figure 7—video 1.** Morph between ASBT$_{NM(Pan)}$ and ASBT$_{NM(ns)}$ focussed on TM4b.

https://elifesciences.org/articles/89167/figures#fig7video1

with the pantoate as the protein moves to the inward-facing state would trigger the release of the pantoate, rather than resulting in the movement of TM1. In binding the pantoate the position of Thr 112 moves by 2.4 Å (*Figure 7—video 1*). The position of this residue is intriguing because in morphing between the putative outward and inward states of the protein, Thr 112 comes within 2 Å of Met 48 on the panel domain. The interaction between Thr112 and Met48 may, therefore, block the protein from switching conformations. It can be speculated that the movement of the threonine side-chain may unlock the transporter, allowing it to switch from outward- to inward-facing (*Figure 7*).

The pantoate binding mode seen for ASBT$_{NM}$ can easily be extrapolated to the plant BASS transporters, which transport similar substrates. In addition to BASS1, which has been shown to transport pantoate in vitro as discussed above (*Huang et al., 2018*), BASS2 transports pyruvate (*Furumoto et al., 2011*), BASS6 glycolate (*South et al., 2017*), and BASS5 chain-elongated 2-keto acids (*Gigolashvili et al., 2009*). The sodium binding sites are conserved throughout these BASS transporters with high conservation within the cross-over regions. Each of these molecules would be able to form hydrogen bonds with the main chain nitrogen atoms of TM4b and TM9b as observed with pantoate. While the pantoate transporter BASS1 contains an asparagine and threonine at the positions of Asn 265 and Thr 112 respectively, in the other BASS transporters these are replaced by serine and glutamine respectively. This may allow the keto-acids to bind. It would be expected, therefore, that a similar mechanism and binding mode may be seen throughout the BASS transporters.

It also seems plausible that in the human bile acid transporters, the substrate would also form specific interactions with the main chain nitrogen atoms of the cross-over helices. As we observe for pantoate binding to ASBT$_{NM}$, for human ASBT it has been shown that uptake of TCH is abolished when the equivalent of Asn 265 (Asn 266) is mutated to a cysteine (*Banerjee et al., 2008*) and increases when Thr 112 (Thr 110 in ASBT$_{NM}$) is changed to the same residue (*Hussainzada et al., 2008*). In NTCP mutation of Asn 262 also abolishes uptake of TCH (*Yan et al., 2014*). This suggests there may be some similarity in the mechanisms but exactly how these proteins are able to bind to the wide variety of primary, secondary, and conjugated bile acids (*Grosser et al., 2021*) that have been reported as substrates is difficult to say. Liu and co-workers have modelled two bile acids into density that they observe when they solved the structure of human NTCP (*Liu et al., 2022*). In this structure, while the glycine head group of the bile acid is near to the cross-over helices, the interactions with it are rather weak and the density associated with this moiety is also rather poorly defined. As the authors of this study have pointed out, it is hard to reconcile this binding mode with the conformational changes associated with an elevator mechanism, which others have demonstrated since that paper was submitted (*Park et al., 2022*). It may well be that the binding mode observed for NTCP is a non-productive mode and a greater interaction with the residues of the cross-over region, linking the sodium ions with substrate binding, will be required to elicit a conformational change. Given that pantoate does not interact with TM1, it seems unlikely that the absence of this helix in the bile acid transporters would affect the elevator mechanism unduly.

In conclusion the elucidation of pantoate-bound ASBT$_{NM}$ provides new insight into the mechanism of the BASS family of transporters. Pantoate binding to the cross-over region of the sodium-bound protein causes subtle changes to Thr 112 and TM4b. Thr 112 is located in the centre of the protein near to the panel domain and its repositioning could unlock the transporter, enabling it to swap between outward- and inward-facing states. In the absence of sodium ions these residues are likely to be more flexible, which may enable the transporter to switch from one conformation to another without requiring the conformational change mediated by the substrate. While the exact binding mode of bile acids in the human proteins remains unclear, the high conservation of residues involved in this area suggests that the interaction with residues on the cross-over region may follow a similar mechanism.

# Materials and methods

**Key resources table**

| Reagent type (species) or resource | Designation | Source or reference | Identifiers | Additional information |
|---|---|---|---|---|
| Gene (*Nesseria meningitidis*) | ASBT$_{NM}$ | https://doi.org/10.1038/nature10450 | ASBT$_{NM}$ | |

*Continued on next page*

| Reagent type (species) or resource | Designation | Source or reference | Identifiers | Additional information |
|---|---|---|---|---|
| | | *Continued* | | |
| Recombinant DNA reagent | PWaldo GFPd-3C (plasmid) | *Hatton et al., 2022* | | Modified from original PWaldo GFPd vector https://doi.org/10.1110/ps.051466205 |
| Strain, strain background (*Escherichia coli*) | Lemo21(DE3) | New England Biolabs | | |
| Chemical compound | Pantoate | Merck Life Science UK | (R)-Pantoic acid sodium salt | |

## Expression and purification

ASBT$_{NM}$ (*Hu et al., 2011*) was subcloned into a modified version of the expression vector, pWaldo GFPd (*Drew et al., 2006*) in which the TEV protease site had been altered to a 3C protease recognition site (*Hatton et al., 2022*). Site-directed mutations were introduced by PCR (Quikchange II, Agilent Technologies). Cultures were grown in Lemo21 (DE3) cells in PASM-5052 media following the MemStar protocol (*Lee et al., 2014*). Briefly, the cells were grown at 37°C with shaking at 200 rpm. At an OD$_{600}$ of 0.5, 0.4 mM IPTG and 0.25 mM L-rhamnose were added and the temperature was decreased to 25°C for overnight induction. Cell pellets were harvested by centrifugation at 5000 × *g* for 15 min at 4°C and resuspended in PBS (137 mM NaCl, 2.7 mM KCl, 10 mM Na$_2$HPO$_4$, 1.8 mM KH$_2$PO$_4$) with 1 mM MgCl$_2$, DNaseI, and 0.5 mM Pefabloc (Roche). Cells were lysed by passing them twice through a cell disruptor at a pressure of 25 kpsi. Unbroken cells and cell debris were pelleted by centrifugation at 15,000 × *g* for 13 min and the supernatant was subjected to ultracentrifugation at 200,000 × *g* at 4°C for 1 hr to pellet the membranes. Membrane pellets were resuspended in PBS, 15 ml per 1 l of culture, snap frozen in liquid nitrogen, and then stored at –80°C.

Membranes were solublised in 1× PBS, 150 mM NaCl, 10 mM imidazole, and 1% (wt/vol) DDM supplemented with 0.5 mM Pefabloc (Roche) for 2 hr at 4°C. Insolubilised material was removed by centrifugation at 200,000 × *g* for 45 min and the supernatant was added to HisPur Ni-NTA superflow agarose, (Thermo Fisher) (1 ml per 1 mg GFP-tagged protein). The slurry was gently stirred for 3 hr to allow binding and then loaded into a glass Econo-Column (Bio-Rad). The column was washed with 10 column volumes (CV) of wash buffer (1× PBS, 150 mM NaCl, 0.1% DDM) containing 20 mM imidazole, followed by 10 CV with the imidazole augmented to 30 mM. 3C protease (1:1 stoichiometry with ASBT$_{NM}$-GFP) was added to the resin and cleavage was performed overnight at 4°C. The protein was eluted with 20 mM Tris-HCl (pH 7.5), 150 mM NaCl, 0.03% DDM, and passed over a 5 ml HisTrap HP column (GE Healthcare) equilibrated with the same buffer. The flow-through was collected and concentrated to 6–10 mg/ml using a 100 kDa molecular weight cutoff centrifugal concentrator (Sartorius) and loaded onto a Superdex 200 Increase 10/300 GL column equilibrated with 20 mM Tris-HCl (pH 7.5), 150 mM NaCl, 0.03% DDM. Fractions containing protein were pooled together and concentrated to ~25 mg/ml as above.

## Protein crystallisation and structure solution

### ASBT$_{NM(Pan)}$

Crystals were grown using the lipidic cubic phase method (*Caffrey and Cherezov, 2009*). The protein was mixed with monoolein at 60:40 (wt/wt) ratio using a coupled syringe device (SPT Labtech) and crystallisation trials were set up at 20°C using glass sandwich plates using a Mosquito Robot (SPT Labtech). The protein was preincubated with 1 mM pantoate for 30 min at room temperature. Crystals appeared after 1 week. Crystals were harvested from MemGold2 (Molecular Dimensions) condition A1, (0.2 M magnesium chloride hexahydrate, 0.005 M cadmium chloride hemi-(pentahydrate), 0.1 M Tris [pH 7.5], and 14% vol/vol PEG 500 MME). Crystals were harvested into MicroMounts (MiTeGen) and snap-cooled in liquid nitrogen. X-ray diffraction data were collected at I24 at Diamond Light Source. Diffraction images were integrated and scaled using DIALS (*Waterman et al., 2016*) with further processing in CCP4 (*Collaborative Computational Project, Number 4, 1994*). The structures were solved by molecular replacement in Phaser (*McCoy et al., 2007*) through the Phenix suite of programs (*Adams et al., 2010*) from a model derived from the deposited structure of ASBT$_{NM}$ (3zuy)

that had been crystallised by vapour diffusion (*Hu et al., 2011*). Refinement was performed in Phenix. refine (*Afonine et al., 2012*) interspersed with manual rebuilding in Coot (*Emsley and Cowtan, 2004*). Pantoate and sodium ions were built into clear density in the maps. Lipids and metal ions were tentatively assigned to other features in these maps. Given that both structures contained metal ions from the crystallisation or purification the structures were refined against $I^+/I^-$.

### ASBT$_{NM(ns)}$

Using the in meso method of crystallisation as above, several structures were solved where TCH was not added to the crystallisation mixture. The highest resolution data were obtained from a single crystal harvested from condition C3 of the MemMeso screen (Molecular Dimensions) with 0.1 M sodium chloride 0.1 M HEPES 7, 30 % vol/vol PEG 300, and 0.1 M calcium chloride dihydrate. The drop also contained (4R-cis)-1-[4-[4-[3,3-dibutyl-7-(dimethylamino)-2,3,4,5-tetrahydro-4-hydroxy-1,1-dioxido-1-benzothiepin-5-yl]-phenoxy]butyl]-4-aza-1-azoniabicyclo[2.2.2]octane methanesulfonate, dissolved in dimethyl sulphoxide (DMSO). This is an inhibitor of human ASBT (*Huang et al., 2005*) that did not show any effect in our stability assays with ASBT$_{NM}$. The data were processed as above. As the ASBT inhibitor could not be observed in the electron density maps and the resultant structure was consistent with lower resolution structures where this compound was not added we treat this as a good representative of the non-substrate-bound structures. Density present in the cavity was modelled as monoolein (*Figure 4—figure supplement 1B*).

Superpositions were performed in Chimera (*Pettersen et al., 2004*) and structural images were prepared in PyMol (*Delano, 2002*). Images involving electron density were prepared in CCP4mg (*McNicholas et al., 2011*). Movies were made with Chimera.

## Stability assay

Screening of compounds for binding was carried out using a stability assay based on binding of 7-diethylamino-3-(4'-maleimidylphenyl)-4-methylcoumarin (CPM) to the protein (*Alexandrov et al., 2008*; *Sonoda et al., 2011*). CPM (Thermo Fisher) was dissolved in DMSO to a final concentration of 4 mg/ ml. The assay was performed in 0.2 ml non-skirted low profile 96-well PCR plates (Thermo Fisher). 50 µl of protein (2.5 µg in 20 mM Tris-HCl [pH 7.5], 150 mM NaCl, 0.03% DDM) was added to each well supplemented with 1 µl (final concentration 1 mM) of each of the compounds of interest and the plate incubated for 30 min at room temperature. The CPM dye was diluted 1:100 in 20 mM Tris-HCl (pH 7.5), 150 mM NaCl, 0.03% DDM, and 2.5 µl of the diluted dye was added to each well. The assay was performed using a Stratagene Mx3005P Real-Time PCR machine (Strategene) and samples were heated from 25°C to 95°C in 1°C/min steps. Data were analysed using GraphPad Prism.

## Isothermal calorimetry

The protein sample was dialysed overnight at 4°C against 20 mM Tris-HCl (pH 7.5), 150 mM NaCl, and 0.03% DDM and centrifuged at 16,000 × *g* at 4°C for 30 min. ITC experiments were performed on a MicroCal PEAQ-ITC (Malvern Panalytical, UK). The protein solution (220 µM) was filled into the sample cell and the pantoate solution (5 mM in the dialysis buffer) into the syringe. The cell temperature was set to 10°C with a stirring speed of 750 rpm and a reference power of 10 µcal/s. 20 injections were performed with an initial delay of 250 s. The initial injection was performed for 0.8 s with an injection volume of 0.4 µl. The later injections were performed for 4 s with an injection volume of 2 µl. 180 s spacing was left between each injection. The data were analysed using the 'one set of sites' model within the MicroCal PEAQ-ITC software (Malvern) iterated using the Lavenberg-Marquardt algorithm after subtraction of the control experiment (pantoate titrated into buffer). The thermodynamic and binding parameters were derived from the nonlinear least squares fit to the binding isotherm.

## MD simulations

The pantoate-bound ASBT structure was embedded in in a 80:20 POPE:POPG bilayer and solvated with neutralising ions (0.15 M NaCl) to a final box size of $9.1 \times 9.1 \times 9.6$ nm$^3$ using CHARMM-GUI (*Jo et al., 2008*; *Lee et al., 2016*; *Wu et al., 2014*). An initial structure was generated without bound sodium. Two sodium ions were moved back to the binding sites manually to generate a sodium-bound initial structure.

From each starting structure, simulations were performed using Gromacs 2018.6 (*Abraham et al., 2015*) with the CHARMM-36 forcefield (*Best et al., 2012*) and TIP3 water. Parameters for pantoate were generated using CGenFF (*Vanommeslaeghe et al., 2010*) and converted to a Gromacs format using the cgenff_charmm2gmx.py script. Energy minimisation and 5 ns multi-step equilibration were performed following the CHARMM-GUI protocol, followed by three 500 ns production runs using different initial velocities. The simulation timestep was set to 2 fs; temperature and pressure were maintained using the stochastic velocity rescaling thermostat (*Bussi et al., 2007*) (at 303.15 K) and the Parrinello-Rahman semi-isotropic barostat (*Parrinello and Rahman, 1981*) (at 1 atm), respectively. The particle-mesh Ewald method (*Darden et al., 1993*) was used for long-range electrostatic interactions, and non-bonded interactions were reduced from 1 nm to a 1.2 nm cutoff using potential shift. Bonds involving hydrogens in ASBT, lipids and pantoate were constrained using the LINCS algorithm (*Hess et al., 1997*); all bonds in the rigid TIP3 water molecules were constrained with SETTLE (*Miyamoto and Kollman, 1992*). Three repeats of 500 ns were carried out for both the structures with no sodium bound (-Na$^+$) and sodium ions bound in both the Na1 and Na2 sites (+Na$^+$). All simulation analysis was performed using MDAnalysis (*Gowers et al., 2016*; *Michaud-Agrawal et al., 2011*). For hydrogen bond analysis, a 3.5 Å distance and 145° angle cutoff were used. Visualisations of structures were made using VMD (*Humphrey et al., 1996*).

## Acknowledgements

We thank the staff at beamline I24 at DLS and the technical support in the School of Life Sciences, University of Warwick.

## Additional information

### Competing interests

Raul Pacheco-Gomez: employee of Malvern Panalytical Ltd. The other authors declare that no competing interests exist.

### Funding

| Funder | Grant reference number | Author |
|---|---|---|
| National Institutes of Health | R01GM118772 | Patrick Becker<br>Fiona Naughton<br>Oliver Beckstein |
| Medical Research Council | MR/P010393/1 | Deborah Brotherton |

The funders had no role in study design, data collection and interpretation, or the decision to submit the work for publication.

### Author contributions

Patrick Becker, Fiona Naughton, Validation, Investigation, Methodology, Writing – review and editing; Deborah Brotherton, Supervision, Investigation, Methodology, Writing – review and editing; Raul Pacheco-Gomez, Investigation, Methodology, Writing – review and editing; Oliver Beckstein, Supervision, Funding acquisition, Validation, Investigation, Writing – review and editing; Alexander D Cameron, Conceptualization, Supervision, Funding acquisition, Validation, Investigation, Methodology, Writing – original draft

### Author ORCIDs

Oliver Beckstein http://orcid.org/0000-0003-1340-0831
Alexander D Cameron https://orcid.org/0000-0001-8776-3518

Reviewer #1 (Public Review): https://doi.org/10.7554/eLife.89167.3.sa1
Reviewer #2 (Public Review): https://doi.org/10.7554/eLife.89167.3.sa2
Reviewer #3 (Public Review): https://doi.org/10.7554/eLife.89167.3.sa3
Author Response https://doi.org/10.7554/eLife.89167.3.sa4

## Additional files

### Supplementary files
• MDAR checklist

### Data availability
Data and coordinates have been deposited in the RCSB Protein Data Bank under accession numbers 8OYG (ASBTNM(Pan)) and 8OYF (ASBTNM(ns)). MD trajectories in the GROMACS XTC format were deposited in the OSF.io repository under DOI 10.17605/OSF.IO/KFDT5 under the open CC-BY Attribution 4.0 International license.

The following datasets were generated:

| Author(s) | Year | Dataset title | Dataset URL | Database and Identifier |
| --- | --- | --- | --- | --- |
| Becker P, Cameron AD | 2023 | Crystal Structure of ASBTNM in complex with pantoate | https://www.rcsb.org/structure/8OYG | RCSB Protein Data Bank, 8OYG |
| Becker P, Cameron AD | 2023 | Crystal structure of ASBTNM in lipidic cubic phase without substrate bound | https://www.rcsb.org/structure/8OYF | RCSB Protein Data Bank, 8OYF |
| Beckstein O, Naughton F | 2023 | MD simulations of the ASBT transmembrane transporter protein | https://doi.org/10.17605/OSF.IO/KFDT5 | Open Science Framework, 10.17605/OSF.IO/KFDT5 |

The following previously published datasets were used:

| Author(s) | Year | Dataset title | Dataset URL | Database and Identifier |
| --- | --- | --- | --- | --- |
| Hu N-J, Iwata S, Cameron AD, Drew D | 2011 | Crystal structure of a bacterial homologue of the bile acid sodium symporter ASBT | https://www.rcsb.org/structure/3ZUX | RCSB Protein Data Bank, 3ZUX |
| Zhou X, Levin EJ, Zhou M | 2013 | Crystal Structure of the sodium bile acid symporter from Yersinia frederiksenii | https://www.rcsb.org/structure/4N7W | RCSB Protein Data Bank, 4N7W |
| Zhou X, Levin EJ, Zhou M | 2013 | The E254A mutant of the sodium bile acid symporter from Yersinia frederiksenii | https://www.rcsb.org/structure/4N7X | RCSB Protein Data Bank, 4N7X |

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
