## [Editor Report · eLife assessment]

The manuscript represents an **important** contribution to an ongoing discussion about the substrate binding site and mechanism of the Bile Acid Sodium Symporter (BASS) family of transporters. Structural and biochemical analysis of a bacterial homolog, ASTBnm, in complex with its native substrate (not bile acids, but a vitamin A precursor, pantoate) show a new binding site that is consistent with classical proposals for elevator-type transport mechanisms. Molecular dynamics (MD) simulations highlight the improved stability for the substrate in the active site when ions are present, suggesting a binding order during the transport cycle. The structural studies, binding assays, and MD simulations are **convincing**.

---

## [Referee Report · Reviewer #1 (Public Review)]

The current manuscript provides a timely contribution to the ongoing discussion about the mechanism of the apical sodium/bile acid transporter (ASBT) transporters. Recent structures of the mammalian ASBT transporters exhibited a substrate binding mode with few interactions with the core domain (classically associated with substrate binding), prompting an unusual proposal for the transport mechanism. Early structures of ASBT homologues from bacteria also exhibit unusual substrate binding in which the core substrate binding domain is less engaged than expected. Due to the ongoing questions of how substrate binding and mechanism are linked in these transporters, the authors set out to deepen our understanding of a model ABST homolog from bacteria N. meningitidis (ABST-NM).

The premise of the current paper is that the bacterial ASBT homologs are probably not physiological bile acid transporters, and that structural elucidation of a natively transported substrate might provide better mechanistic information. In the current manuscript, the authors revisit the first BASS homologue to be structurally characterized, ABST-NM. Based on bacteriological assays in the literature, the authors identify the coenzyme A precursor pantoate as a more likely substrate for ABST-NM than taurocholate, the substrate in the original structure. A structure of ASBT-NM with pantoate exhibits interesting differences in structure. The structures are complemented with MD simulations, and the authors propose that the structures are consistent with a classical elevator transport mechanism.

The structural experiments are convincing. The binding and molecular dynamics experiments provide intriguing insights into the transporter's conformational changes. However, it is nonetheless a soft spot in the story that a transport assay is not readily available for this substrate. Mechanistic proposals, like the proposed role of T112 in unlocking the transporter, would be better supported by transport data.

---

## [Referee Report · Reviewer #2 (Public Review)]

The manuscript starts with a demonstration of pantoate binding to ASBTnm using a thermostability assay and ITC, and follows with structure determinations of ASBTnm with or without pantoate. The structure of ASBTnm in the presence of pantoate pinpoints the binding site of pantoate to the "crossover" region formed by partially unwinded helices TMs 4 and 9. Binding of pantoate induces modest movements of side chain and backbone atoms at the crossover region that are consistent with providing coordination of the substrate. The structures also show movement of TM1 that opens the substrate binding site to the cytosol and mobility of loops between the TMs. MD simulations of the ASBT structure embedded in lipid bilayer suggests a stabilizing effect of the two sodium ions that are known to co-transport with the substrate. Binding study on pantoate analogs further demonstrate the specificity of pantoate as a substrate.

Overall, the structural, functional and computational studies are solid and rigorous, and the conclusions are well justified. In addition, the authors discussed the significance of the current study in a broader perspective relevant to recent structures of mammalian BASS members.

---

## [Referee Report · Reviewer #3 (Public Review)]

The manuscript describes new ligand-bound structures within the larger bile acid sodium symporter family (BASS). This is the primary advance in the manuscript, together with molecular simulations describing how sodium and the bile acids sit in the structure when thermalized. What I think is fairly clear is that the ligands are more stable when the sodiums are present, with a marked reduction in RMSD over the course of repeated trajectories. This would be consistent with a transport model where sodium ions bind first, and then the bile acid binds, followed by a conformational change to another state where the ligands unbind.

While the authors mention that BASS transporters are thought to undergo an elevator transport mechanisms, this is not tested here. In my reading, all the crystal structures belong to the same conformational state in the overall transport cycle, and the simulations do not make an attempt to induce a transition on accessible simulation timescales. Instead, there is a morph between two inward facing states.

The focus is on what kinds of substrates bind to this transporter, interrogating this with isothermal calorimetry together with mutations. With a Kd in the micromolar range, even the best binder, pantoate, actually isn't a particularly tight binder in the pharmaceutical sense. For a transporter, tight binding is not actually desirable, since the substrate needs to be able to leave after conformational change places it in a position accessible to the other side.

The structure and simulation analysis falls into the mainstream of modern structural biology work.

---

## [Author Response]

The following is the authors’ response to the original reviews.

**Reviewer #1 (Public Review):**
The current manuscript provides a timely contribution to the ongoing discussion about the mechanism of the apical sodium/bile acid transporter (ASBT) transporters. Recent structures of the mammalian ASBT transporters exhibited a substrate binding mode with few interactions with the core domain (classically associated with substrate binding), prompting an unusual proposal for the transport mechanism. Early structures of ASBT homologues from bacteria also exhibit unusual substrate binding in which the core substrate binding domain is less engaged than expected. Due to the ongoing questions of how substrate binding and mechanism are linked in these transporters, the authors set out to deepen our understanding of a model ABST homolog from bacteria N. meningitidis (ABST-NM).The premise of the current paper is that the bacterial ASBT homologs are probably not physiological bile acid transporters, and that structural elucidation of a natively transported substrate might provide better mechanistic information. In the current manuscript, the authors revisit the first BASS homologue to be structurally characterized, ABST-NM. Based on bacteriological assays in the literature, the authors identify the coenzyme A precursor pantoate as a more likely substrate for ABSTNM than taurocholate, the substrate in the original structure. A structure of ASBT-NM with pantoate exhibits interesting differences in structure. The structures are complemented with MD simulations, and the authors propose that the structures are consistent with a classical elevator transport mechanism.The structural experiments are generally solid, although showing omit maps would bolster the identification of the substrate binding site.

We have added an omit map in Fig S2.

One shortcoming is that, although pantoate binding is observed, the authors do not show transport of this substrate, undercutting the argument that the pantoate structure represents binding of a "better" or more native substrate. Mechanistic proposals, like the proposed role of T112 in unlocking the transporter, would be much better supported by transport data.

In the absence of being able to source radiolabelled pantoate at a reasonable cost, we decided to focus on binding studies, relying on the fact that pantoate/pyruvate uptake has been shown in other BASS transporters. While we agree that transport needs to be substantiated, our crystallographic and molecular dynamics studies combined provide a picture of sodium ions stabilising the substrate binding site to enable the binding of the substrate, which in turn induces further conformational changes. Such changes would be consistent with a mechanism of sodium driven transport with clear coupling of the sodium ions to substrate translocation. We are not saying this is a “better” substrate but rather that a substrate binding like this would be able to elicit the conformational changes necessary for transport – something that has been missing from previous studies.

**Reviewer #2 (Public Review):**
The manuscript starts with a demonstration of pantoate binding to ASBTnm using a thermostability assay and ITC, and follows with structure determinations of ASBTnm with or without pantoate. The structure of ASBTnm in the presence of pantoate pinpoints the binding site of pantoate to the"crossover" region formed by partially unwinded helices TMs 4 and 9. Binding of pantoate induces modest movements of side chain and backbone atoms at the crossover region that are consistent with providing coordination of the substrate. The structures also show movement of TM1 that opens the substrate binding site to the cytosol and mobility of loops between the TMs. MD simulations of the ASBT structure embedded in lipid bilayer suggests a stabilizing effect of the two sodium ions that are known to co-transport with the substrate. Binding study on pantoate analogs further demonstrates the specificity of pantoate as a substrate.The weakness of the manuscript includes a lack of transport assay for pantoate and a lack of demonstration that the observed conformational changes in TM1 and the loops are relevant to the binding or transport of pantoate.

We agree that the manuscript would have been bolstered by transport data (see response to reviewer 1). The take-home message from the movement of TM1 and the loops is that they are flexible. It is probably unlikely that TM1 moves like this during the transport cycle and we have avoided overplaying the significance of this movement. Instead, we have focussed on the conformational changes in the pantoate binding site. We have made an additional movie concentrating on the binding site and not including TM1.

Overall, the structural, functional and computational studies are solid and rigorous, and the conclusions are well justified. In addition, the authors discussed the significance of the current study in a broader perspective relevant to recent structures of mammalian BASS members.
**Reviewer #3 (Public Review)**
The manuscript describes new ligand-bound structures within the larger bile acid sodium symporter family (BASS). This is the primary advance in the manuscript, together with molecular simulations describing how sodium and the bile acids sit in the structure when thermalized. What I think is fairly clear is that the ligands are more stable when the sodiums are present, with a marked reduction in RMSD over the course of repeated trajectories. This would be consistent with a transport model where sodium ions bind first, and then the bile acid binds, followed by a conformational change to another state where the ligands unbind.While the authors mention that BASS transporters are thought to undergo an elevator transport mechanisms, this is not tested here. In my reading, all the crystal structures describe the same conformational state, and the simulations do not make an attempt to induce a transition on accessible simulation timescales. Instead, there is a morph between two states where different substrates are bound, which induces a conformational change that looks unrelated to the transport cycle.

To make our conclusions clearer we have added another movie showing a morph between the structure without substrate (instead of using the structure with taurocholate, which we were using as a representative of the unbound structure) and that with pantoate and have omitted the panel domain including TM1. While both of these structures are inward-facing, there are significant conformational changes within TM4 that we have described in the article.

Instead, the focus is on what kinds of substrates bind to this transporter, interrogating this with isothermal calorimetry together with mutations. With a Kd in the micromolar range, even the best binder, pantoate, actually isn't a particularly tight binder in the pharmaceutical sense. For a transporter, tight binding is not actually desirable, since the substrate needs to be able to leave after conformational change places it in a position accessible to the other side.

As the referee points out the Kd that we observe would be consistent with those for substrates of other transporters.

There is one really important point that readers and authors should be aware of. In Figure 2A, the names are not consistent with the chemical structure. "-ate" denotes when a carboxylic acid is in the deprotonated form, creating a charged carboxylate. What is drawn is pantoic acid, ketopantoic acid, and pantoethenic acid. Less importantly, the wedges and hashes for the methyl group are arguably not appropriate, since the carbon they are attached to is not a chiral center. For the crystallization, this makes no difference, since under near-neutral pKas the carboxylic acid will spontaneously deprotonate, and the carboxylate form will be the most common. However, if the structures in Figure 2A were used for classical molecular simulation, that would be a big problem, since now that would be modeling the much rarer neutral form rather than the charged state. I am reasonably sure based on Figure 5 that the MD correctly modeled the deprotonated form with a carboxylate, but that is inconsistent with Figure 2A. Otherwise, the structure and simulation analysis falls into the mainstream of modern structural biology work.

We have corrected the inconsistency of the protonaNon state in the naming of the molecular structures. Thank you for poinNng this out – though the names represented the predominant form in soluNon, the more aestheNcally pleasing protonated form got the beOer of us in our representaNons. The correct form was used in the MD.

**Reviewer #1 (Recommendations For The Authors):**
1. Omit maps (Fo-Fc) should be shown for pantoate and for the sodiums in the structure.

This has been added to supplementary Figure 2.

1. Line 86 - could you briefly describe the alternative mechanism proposed for the mammalian NTPCs?

We have added an extra line to describe this deviation from the classical alternating access model.

1. Line 124 - where is the lipid like molecule, and does it interact with either the kinked helix or the substrate? A supplemental figure would be helpful.

The lipid like molecule lies between the substrate and the kinked helix, but doesn’t interact strongly with either. It would appear that the lipid would bind in the crevice rather than causing the crevice. We add Author response image 1 here but have not added it to the supplementary figures. The maps and PDB file are available for download.

**Author response image 1. sa4fig1:** The 2mFo-DFc density is at 1σ, the mFo-DFc density is at 2. 5σ.

1. I notice that the apo and pantoate structures are crystallized in different space groups. How does this compare to the original TCH structure? Is there any chance that crystal packing is altering the TM1 geometry or loop 1?

We cannot rule out the effect of the crystallisation conditions on the movement of the TM1. We have now solved a number of different structures of ASBTNM and this is the first time we observe TM1 in this conformation. As stated above we have refrained from overplaying the significance of the movement of TM1 to transport, other than to say that some adjustments need to be made to accommodate the pantoate.

**Reviewer #2 (Recommendations For The Authors):**
Minor comments:Pg 3, "... with a 5-fold inverted repeat...", Should be 2-fold?

Changed, thank you.

**Reviewer #3 (Recommendations For The Authors):**
Is there any chance that the MD simulations (even in a reduced form) could be uploaded to Zenodo or a similar repository?

We have taken up this suggestion and added the information in the paper: MD trajectories in the GROMACS XTC format were deposited in the OSF.io repository under DOI 10.17605/OSF.IO/KFDT5 under the open CC-BY Attribution 4.0 International license. The trajectories contain all atoms and were subsampled at 5-ns intervals. GROMACS run input files (TPR format) and initial coordinate files (GRO format) together with topology files (GROMACS format) are also included.

Watch the "Å" symbol in Figures 5, S6, S7. This looks like they were made in matplotlib, and probably used something like: "$\AA$", which puts the symbol in math mode. This makes the Å symbol in italics. Matplotlib has gotten better UTF-8 support

Changed, thank you.

Your citation for LINCS duplicates the citation for PME. I think you want the Hess 1998 paper.10.1002/(SICI)1096-987X(199709)18%3A12<1463%3A%3AAID-JCC4>3.0.CO%3B2-H

Changed, thank you